# Complete Chloroplast Genomes and Comparative Analyses of *L. chinensis*, *L. anhuiensis*, and *L. aurea* (Amaryllidaceae)

**DOI:** 10.3390/ijms21165729

**Published:** 2020-08-10

**Authors:** Fengjiao Zhang, Tao Wang, Xiaochun Shu, Ning Wang, Weibing Zhuang, Zhong Wang

**Affiliations:** 1Jiangsu Key Laboratory for the Research and Utilization of Plant Resources, Institute of Botany, Jiangsu Province and Chinese Academy of Sciences (Nanjing Botanical Garden Mem. Sun Yat-Sen), Nanjing 210014, China; johnwt1007@163.com (T.W.); islbe@163.com (X.S.); wangning813@njau.edu.cn (N.W.); weibingzhuangnj@sina.com (W.Z.); 2The Jiangsu Provincial Platform for Conservation and Utilization of Agricultural Germplasm, Institute of Botany, Jiangsu Province and Chinese Academy of Sciences (Nanjing Botanical Garden Mem. Sun Yat-Sen), Nanjing 210014, China

**Keywords:** *Lycoris*, chloroplast genome, comparative analysis, phylogeny

## Abstract

The genus *Lycoris* (about 20 species) includes important medicinal and ornamental plants. Due to the similar morphological features and insufficient genomic resources, germplasm identification and molecular phylogeny analysis are very limited. Here, we sequenced the complete chloroplast genomes of *L. chinensis*, *L. anhuiensis*, and *L. aurea*; they have very similar morphological traits that make it difficult to identify. The full length of their cp genomes was nearly 158k bp with the same guanine-cytosine content of 37.8%. A total of 137 genes were annotated, including 87 protein-coding genes, 42 tRNAs, and eight rRNAs. A comparative analysis revealed the conservation in sequence size, GC content, and gene content. Some variations were observed in repeat structures, gene expansion on the IR-SC (Inverted Repeat-Single-Copy) boundary regions. Together with the cpSSR (chloroplast simple sequence repeats), these genetic variations are useful to develop molecular markers for germplasm identification. Phylogenetic analysis showed that seven *Lycoris* species were clustered into a monophyletic group, and closed to *Narcissus* in Amaryllidaceae. *L. chinensis*, *L. anhuiensis*, and *L. longituba* were clustered together, suggesting that they were very likely to be derived from one species, and had the same ancestor with *L. squamigera*. Our results provided information on the study of genetic diversity, origins or relatedness of native species, and the identification of cultivars.

## 1. Introduction

The monocot genus *Lycoris* (Amaryllidaceae) is a herbaceous perennial species and commonly used as an important medicinal and ornamental plant. It is rich in Amaryllidaceae-type alkaloids that have been used in immunostimulatory, anti-malarial, tumor-suppressing, and viral activity studies [1,2]. It also has been used as a garden flower due to the colorful and special flowers; another feature is the absence of leaves while flowering, so it is also called the lily [3]. *Lycoris* is native to eastern Asia and confined to temperate China, Japan, and Korea [4].

*Lycoris* consists of about 20 species, of which 15 species and one variety are distributed in China according to the record of “Flora of China”, which represents the most comprehensive catalog. However, several new species continue to be reported, such as *L. hunanensis* [5], *L. hubeiensis* [6], and some hybrids, suggesting that more species are present in the genus than we now recognize. Highly frequent natural hybridization caused the wide morphological variation in the genus, so there was a general assumption that *Lycoris* is of hybrid origin [7], and *L. radiata* was considered to be a possible ancestor of the genus *Lycori*, according to the phylogenetic analysis using chloroplast genome regions, but it is not consistent with the karyotype evolution study [8]. The complex genetic background makes it difficult to identify some species with high similarity on morphology and clarify the phylogenetic analysis. In the past years, to understand the genetic diversity, possible origin species, maternal donors, and an evolutionary relationship of *Lycoris*, morphological characters [9], plastid DNA sequences (such as *matk*, *rbcL*, *atpB*-*rbcL* IGS, *trnL*-*F*, *trnS-trnfM,* and *trnC-ycf6*) [8,10,11] and molecular markers, such as random amplification of polymorphic DNA (RAPD) markers [12], inter-simple sequence repeat (ISSR) markers [13], and Start codon targeted polymorphism (SCoT) markers [14] were reported. However, compared to the results of each method, some results were conflicting, suggesting that these selected loci and fragments were not sufficient for the phylogenetic information to elucidate the evolutionary relationships among *Lycoris* species.

In the past decade, the technology of Next Generation Sequencing (NGS) and the application of DNA barcodes have helped to improve our understanding of the species and relationships in many groups of plants [15,16]. In eukaryotes, the chloroplast is the core organelle for photosynthesis and carbon fixation [17]. Their genome sequences are becoming a very useful tool in identifying closely related plant species, and have been widely used for plant evolution and phylogeny [18,19,20]. The complete chloroplast genome information has enhanced our understanding of plant biology, diversity, and evolutionary relationships. Moreover, it could reveal considerable variation within species in terms of both sequence and structural variation. It has been widely used in many genera and family analysis, such as *Prunus* [21], *Aristolochia* [22], and Araceae [23]. Therefore, a more species’ complete chloroplast genome will offer especially valuable information for species identification and conservation of valuable traits, then facilitate the breeding of closely related species and phylogenetic analysis [24]. To date, five *Lycoris* chloroplast genomes have been reported, with published whole sequences and simple clustering but there have been no reports on the structure and comparative analysis of the cp genomes, which need to be paid more attention to analyze their difference by interspecific comparison.

Here, we sequenced three complete chloroplast genomes of *Lycoris* species, *L. chinensis*, *L. anhuiensis,* and *L. aurea*, which have highly similar flowers and are difficult to identify by leaf and bulb morphology. To explore their genetic divergence and phylogenetic relationships, we compared the different characteristics of the cp genomes of these three species and other four reported *Lycoris* species, they were *L. squamigera* [25], *L. radiata* [26], *L. longitube* [27], and *L. sprengeri* [28]. An interspecific comparative analyses of the cp genomes provide valuable molecular markers for germplasm identification, and the phylogenetic analysis may significantly improve the low resolution in evolutionary relationships and contribute to the conservation, domestication, and utilization of *Lycoris* plants.

## 2. Results and Discussion

### 2.1. General Features of the Lycoris Chloroplast Genomes

Using next-generation sequencing technology, the total DNA of *L. chinensis*, *L. anhuiensis,* and *L. aurea* were sequenced. As a result, a total of 14 million, 8 million, and 7 million paired-end reads of 150 bp were used for chloroplast genome assembly, and the average organelle coverage reached 14,255×, 8624×, and 7497× sequencing depth, respectively. The organelle assembly NOVOPlasty Version 3.3 [29] was conducted to de novo assemble the cp genomes where the sizes of *L. chinensis*, *L. anhuiensis*, and *L. aurea* were 158,484 bp, 158,490 bp, and 158,367 bp, respectively (Table 1). The other four reported species are also presented in Table 1. The length of *L. radiata* and *L. aurea* was close, *L. sprengeri* was the longest, and the other four species were ranging from 158,480 to 158,490 bp. Their GC contents were 37.8%. Compared with *Allium* (Amaryllidaceae) [30], most of their genome size was between 152k bp and 155k bp, the GC content varied from 36.7% to 37.0%. Here, longer cp genome sequences and higher GC content were observed in *Lycoris* species. They exhibited the typical quadripartite structure, consisting of a large single-copy region (LSC) (86,458–86,612 bp), a small single-copy region (SSC) (16,620–18,540 bp), and a pair of inverted repeats (IRs) (26,731–27,568 bp) (Table 1 and Figure 1).

In seven *Lycoris* species, a total of 137 genes were identified, including 87 protein-coding genes (PCGs), 42 tRNA genes, and eight rRNA genes, of these, 20 genes were duplicated in the different regions (Table 1 and Table 2), suggesting the high conservation of cp genes in *Lycoris*. In the three species we provided in this study, the gene distribution in *L. aurea* was slightly different from *L. chinensis* and *L. anhuiensis*. In *L. aurea*, the LSC region contained 63 protein-coding genes (*rps12* duplicated) and 21 tRNA genes, of these genes, *rps19* distributed in the LSC and IR regions. In *L. chinensis* and *L. anhuiensis*, 62 PCGs (*rps12* duplicated) and 22 tRNA genes were distributed in the LSC region, which had no *rps19* but one more *trnN*-*GUU*. In three species, the gene distribution in the SSC region was consistent, which contained 13 PCGs (*ycf1* duplicated) and one tRNA, two *ycf1* and *ndhF* spanned the SSC and IR regions (Figure 1). The comparative analysis showed the various number of PCGs and tRNAs in different species, but the rRNA number was conserved, where there were 81–87 PCGs in *Allium* [30], 79 in *Amomum* [31], 80 in *Croomia* and *Stemona* [20], but they all had eight rRNA genes (duplicated four rRNAs in the IR regions).

In most plant chloroplast genomes, Group II (G2) introns comprise the majority of noncoding DNA and could provide rich sequence characters for infrageneric and intrafamilial comparative analysis and phylogeny construction [32,33]. The sole group I intron, *trnL* intron, was not detected in *Lycoris* species, and the *trnL* gene itself was lost in their plastomes. In *Lycoris*, there were 17 same-splitting genes in *L. chinensis*, *L. anhuiensis*, *L. aurea,* and *L. longituba*, 18 in *L. radiata* and *L. sprengeri*, including one more *ndhF*, where the 5′ end exon was located in the IR region and the 3′ end exon and intron were located in the SSR regions. Compared with other *Lycoris* species, there were only 14 splitting genes in *L. squamigera*; however, it contained four specific genes (*trnS*-*CGA*, *rpoC2*, *trnC-ACA,* and *trnE*-*UUC*). In all seven cp genomes, there were nine identical splitting genes, which were *atpF*, *rpoC1*, *ycf3*, *trnL*-*UAA*, *clpP*, *rpl2*, *ndhB*, *trnA*-*UGC*, and *ndhA*; two of them (*ycf3* and *clpP*) contained two introns and three exons, and others consisted of one intron and two exons (Appendix A). In seven *Lycoris* species, the *rps12* intron was lost, which was the same as the three members of Anemone [34]. In *L. squamigera*, the *rpl16* intron was absent, that also occurred in some members of Geraniaceae, Goodeniaceae, and Plumbaginaceae [35,36]. In *Lycoris*, the *rpoC1* intron was located in the LSC region, but it was missing in members of the Poaceae, Aizoaceae, Fabaceae, and in [37,38,39]. The differences of introns between species and conservation in the genus provided information on evolutionary relationships in infrageneric and intergeneric studies.

### 2.2. cpSSRs Analyses and Repeat Structures

Chloroplast simple sequence repeats (cpSSRs) are typically mononucleotide tandem repeat DNA sequences that are widely used as an important molecular marker for ecological and evolutionary processes in plants [40,41]. When there are no effective genomic resources, the simplicity of PCR amplification and the polymorphism of cpSSRs make it easy to use as a marker for characterizing genetic variation. They commonly show intraspecific variation in repeat numbers when located in the noncoding regions of the cp genome [42,43]. Here, we analyzed the number and type of cpSSRs in seven *Lycoris* species. There were 46, 45, 54, 53, 46, 46, and 44 SSRs in the *L. chinensis*, *L. anhuiensis*, *L. aurea*, *L. radiata*, *L. longaituba*, *L. sprengeri*, and *L. squamigera* cp genomes, respectively (Figure 2a and Appendix A). The SSRs were largely distributed in LSC regions (72.7–79.6%), 9–10 SSRs were in the SSC region, and two SSRs in IR regions in seven species that were analyzed in the present study (Figure 2b). For the different unit size, the mononucleotide was the most frequent, accounting more than 95% of all types in the seven species, in which base T and A were the primary elements, one C motif in five species, and one G only in *L. aurea* (Figure 2c). There was one dinucleotide (TA) and no tetranucleotide repetition in seven species, showing the conservation among species. However, the trinucleotide (ATT) was detected in only four species, which were *L. aurea*, *L. radiata*, *L. sprengeri*, and *L. squamigera*, suggesting some variation in different species. Our results showed the intraspecific variation in repeat number, repeats distribution, and repeat motifs, the highly similar morphological characteristics of *L. chinensis* and *L. anhuiensis* presented minor SSRs changes. Compared with other published species, the cpSSR loci number ranged from a few to several hundred, such as nice in *Cryptomeria japonica* [44], 237 in *Magnolia kwangsiensis* [45]. The cpSSRs number of *Lycoris* was between 40 and 60, and the cpSSRs were primarily composed of short polyadenine or polythymine repeats but rarely tandem guanine or cytosine repeats, which was consistent with the previous hypothesis [45]. It is the first report about the SSRs of *Lycoris* complete cp genome, in the absence of genome resources of *Lycoris*, the cpSSRs were available markers for characterizing genetic variation and could offer unique insights into the evolutionary processes of *Lycoris* species and their relatives.

### 2.3. Codon Usage

Codon usage patterns and nucleotide composition of the cp genome could provide a theoretical basis for genetic modification of the chloroplast genome. We deduced the amino acid frequency, the number of codon usage, and the relative synonymous codon usage (RSCU) in the seven *Lycoris* species (Appendix A). Seven species presented the 64 RSCU, and the number of codons ranged from 33,640 to 49,982 in seven species, the least in *L. squamigera*. *L. chinensis* and *L. anhuiensis* had the same patterns of codon usage of numbers and types. Leucine and cysteine were the most and least prevalent amino acids, ATT (encoding isoleucine), and TGA (encoding translational stop) were the most and least used codons in *Lycoris*, respectively. Except for methionine and tryptophan, almost all amino acids had more than one synonymous codon; of these, leucine, arginine and serine had the most (six codon usages). When the RSCU value was >1.00, we defined that it was preferred, and vice versa. There were 32 preferred and 30 non-preferred (RSCU < 1.00) codon usages were identified in five species, which were *L. chinensis*, *L. anhuiensis*, *L. aurea*, *L. radiata*, and *L. longituba*, 31 preferred and 30 non-preferred in *L. sprengeri*, 30 preferred and 31 non-preferred in *L. squamigera* (Appendix A). All seven species had the three same stop codons (TAA, TAG, and TGA). Most of the preferred codon usages end with base A or T, and non-preferred codon usages end with base C or G.

### 2.4. Inverted Repeats Contraction, Expansion, and Interspecific Comparison

The circular structure of chloroplast genome makes four boundaries among IR, LSC and SSC, which are IRB- LSC, IRB-SSC, IRA-SSC, and IRA-LSC. With the evolution of the genome, the shrinkage and expansion of the IR boundary causes the different sizes of the cp genomes and certain genes to enter the IR region or single-copy region [46]. The IR regions of the seven *Lycoris* species are shown in Figure 3, where the gene number and order were conserved, but minor differences were still exhibited at the boundaries. The IR-LSC borders of *L. radiate* and *L. aurea* were within the *rps19*, where some of the sequences were in the LSC region, showing a partial duplication, but the other five species exhibited IR expansion, leading to the entire *rps19* duplication in the IR regions, which is consistent with some species, such as *Vigna radiata* [47] and *Phaseolus vulgaris* [48]. The IR-SSC boundaries of all species are on the *ycf1*, which is common in angiosperms whose complete copy of *ycf1* spanned the SSC-IR regions [49], 925 bp *ycf1* duplicated at the IR regions in four species, which were *L. chinensis*, *L. anhuiensis*, *L. longituba*, and *L. squamigera*. The *trnH* gene is duplicated at the IR regions in all seven species, but there is variation in *trnR* and *trnN* genes, which were not detected at the IRa region in *L. aurea*. By comparing the IRb/SSC and SSC/IRa regions of *ndhF*, and *ycf1*, high similarity was revealed in the *L. chinensis*, *L. anhuiensis*, *L. longituba* and *L. squamigera*, *L. aurea* showed the most obvious difference at IR-SSC boundary (Figure 3).

To reveal the conservation and divergence in *Lycoris* species, interspecific comparisons were performed using online software mVISTA [50,51], which was commonly used to compare the cp genome sequences from two or more species and visualize the alignment results [31,52]. Here, we compared seven species using annotated *L. squamigera* plastome as a reference, including three species we offered in this study and four species previously reported (Figure 4). The alignment revealed high sequence conservation of the seven *Lycoris* species, especially in the IR regions, but non-coding regions showed more divergence than the coding regions. The exons of *ndhF* and *ycf1* were presented the most obvious difference with others, which was the same as the IR boundary analysis.

### 2.5. Phylogenetic Analysis

To explore the phylogenetic positions and evolutionary relationships of *Lycoris* species, we selected 16 species from five families for the construction of maximum likelihood (ML) tree (Figure 5). The complete cp genome sequences and 72 common protein-coding genes were used for analysis, respectively. On the basis of the topologic structure, each of the five selected families (Amaryllidaceae, Asparagaceae, Orchidaceae, Liliaceae, Iridaceae) was clustered into a monophyletic branch respectively. Seven *Lycoris* species formed a monophyletic clade in both trees. They were close to *Narcissus* in the Amaryllidaceae. Our previous analysis using four complete cp sequences presented the close relationship between *L. longituba* and *L. squamigera* [27], here, because of the addition of three species, it showed a closer relationship among the *L. longituba*, *L. chinensis,* and *L. anhuiensis*, but it was still on a subclade with *L. squamigera*. The complete cp genome analysis supported a closer relationship between *L. chinensis* and *L. longituba* (Figure 5a), but the tree constructed by the PCGs showed a closer relationship between *L. chinensis* and *L. anhuiensis* (Figure 5b), which was same with the morphological feature classification. The previous study considered that *L. anhuiensis* might be another variety of *L. longituba* [27]. In the present study, two phylogenetic trees showed a tiny difference in their relationship, suggesting that *L. anhuiensis*, *L. longituba* and *L. chinensis* were very likely to be derived from one species, and had the same ancestor with *L. squamigera*. The flower of *L. aurea* was similar to *L. chinensis*, but it was clustered with *L. radiata*, which shared a similar growth habit. *L. sprengeri* was separated into an independent branch, which was the same as our previous result [27]. Here, the three species with the most similar morphological characteristics were not clustered together, suggesting the inconsistent of phylogenetic relationships in morphology and genomics.

Before the complete cp sequences were published, interspecific relationship and phylogenetic analysis of *Lycoris* was commonly relied on rDNA internal transcribed spacer (ITS) sequences [53,54] and some specific plastid gene sequences, such as *matK*, *atpB*, *rbcL* [8], *trnS*-*trnfM*, and *trnC*-*ycf6* [11]. However, the result of ITS analysis was different from plastid DNA phylogeny, and different plastid fragments also derived differences in results, natural hybridization, and reticulation of *Lycoris* may be the initial cause. Considering the *Lycoris* hybrid origin, maternal inheritance characteristics of chloroplast genome could provide more evidence to clarify their intraspecific evolution. In this study, we sequenced three cp genome of *Lycoris*, and performed the comparative and phylogenetic analyses with four reported sequences, the result revealed the more accurate relationship of these species. Although it was not included all species in the genus, our result provides more cp genome information of *Lycoris*, and makes sure the closest relationship between *L. chinensis, L. anhuiensis* and *L. longituba*.

## 3. Materials and Methods

### 3.1. Sample Collection, DNA Extraction, and Sequencing

*L. chinensis*, *L. anhuiensis* and *L. aurea* were planted in Nanjing Botanical Garden, Mem. Sun Yat-sen (E118_83, N32_06), Nanjing, China. Their specimens were stored at the herbarium of Institute of Botany, Jiangsu Province and Chinese Academy of Sciences with the no. of NAS00591936, NAS00585494, and NAS00585496, respectively. Fresh leaves were collected, quick-freezed in liquid nitrogen, and stored at −80 °C for DNA extraction. Genomic DNA was isolated using the Plant Genomic DNA Kit (Huayueyang, Beijing, China) according to the instructions. DNA integrity was examined by electrophoresis in 1% (*w/v*) agarose gel, the concentration was measured, and quantified accurately using NanoDrop spectrophotometer 2000 (Thermo Scientific; Waltham, MA, USA) and Qubit 2.0 respectively. Finally, high-quality DNA was used for libraries’ construction and sequencing, which was sequenced at Novogene (Tianjin, China) using Nova-PE150 strategy with an insert size of 350 bp for high-throughput sequencing following the manufacturer’s protocol.

### 3.2. Cp Genome Assembly, Annotation and Structure Analysis

The complete cp genomes were assembled by the organelle assembly NOVOPlasty Version 3.3 [29]. The complete cp genome sequence of *L. radiata* (accession no. MN158120) was selected as a reference [26]. Assembled sequences were manually corrected and circularized by BLASTn comparison with the cp genome of *L. radiata*. GC content was analyzed using Geneious software (version R11, http://www.geneious.com). Then, the web server CPGAVAS2 (http://www.herbalgenomics.org/cpgavas2) [55] was used for cp genome annotation and visualization with the default parameters. The relative synonymous codon usage (RSCU) values were calculated using MEGAX [56]. The obtained cp genome of *L. chinensis*, *L. anhuiensis*, and *L. aurea* were deposited in the National Center for Biotechnology Information (NCBI), under the accession numbers of *L. chinensis* (MT700549), *L. anhuiensis* (MT700550) and *L. aurea* (MN158985).

### 3.3. Interspecific Genome Comparison

Four reported complete cp genomes of *Lycoris* were downloaded from the NCBI database, which were *L. squamigera* (MH118290), *L. radiata* (MN158120), *L. sprengeri* (MN158986), and *L. longituba* (MN096601). Chloroplast simple sequence repeat (cpSSR) markers were identified using MISA [43]. The definition of microsatellites (unit size/minimum repeats) was (1/10) (2/6) (3/5) (4/5) (5/5) (6/5) and the maximum length of sequence between two SSRs was set as 100. IRscope (https://irscope.shinyapps.io/irapp/) was used for visualizing the genes’ differences on the boundaries of the junction sites of the seven chloroplast genomes [57]. mVISTA web-interface [50] was performed to align and compare the seven *Lycoris* cp genome sequences. Shuffle-LAGAN was selected as the alignment program for detecting the rearrangements and inversions in sequences.

### 3.4. Phylogenetic Analyses

In this study, we obtained three *Lycoris* cp genome sequences, and 13 complete cp sequences of related species were downloaded from the NCBI database, including four *Lycoris* species, four species in Amaryllidaceae, and five species in the other four families. Their Genbank accession number are shown in Appendix A. Both the complete cp genome sequences and 72 common PCGs were used for tree construction. The 72 common PCGs were *rps12*, *psbA*, *matK*, *psbK*, *psbI*, *atpA*, *atpF*, *atpH*, *atpI*, *rpoC2*, *rpoC1*, *rpoB*, *petN*, *psbM*, *psbD*, *psbC*, *psbZ*, *rps14*, *psaB*, *psaA*, *ycf3*, *rps4*, *ndhJ*, *ndhK*, *ndhC*, *atpE*, *atpB*, *rbcL*, *accD*, *psaI*, *ycf4*, *cemA*, *petA*, *psbJ*, *psbL*, *psbF*, *psbE*, *petL*, *petG*, *psaJ*, *rpl33*, *rps18*, *rpl20*, *clpP*, *psbB*, *psbT*, *psbN*, *psbH*, *petB*, *petD*, *rpoA*, *rpl36*, *rps8*, *rpl14*, *rpl16*, *rps3*, *rpl22*, *rpl2*, *rpl23*, *ycf2*, *ndhB*, *rps7*, *ndhF*, *ccsA*, *psaC*, *ndhE*, *ndhG*, *ndhI*, *ndhA*, *ndhH*, *rps15*, *ycf1*, which were extracted and aligned using MAFFT v7.458 [58,59], and manually verified. Maximum likelihood (ML) phylogenies based on the best-fit model of TVM+F+R2 was conducted using IQ-TREE v. 2.0.3 [60]. The Best-fit model by ModelFinder [61], according to Bayesian information criterion (BIC) and the robustness of the topology, was estimated using 1000 bootstrap replicates.

## 4. Conclusions

In this study, the complete cp genome sequences of *L. chinensis*, *L. anhuiensis*, and *L. aurea* were reported, and interspecific comparison of complete cp genome sequences was analyzed for the first time in the genus *Lycoris*. Compared with the published four cp genome of *Lycoris* species, we found that the genome size, genomic structure, and gene content of the genus *Lycoris* were comparatively conserved, but the IR-SC boundary regions were distinct in seven cp genomes. In all seven species, *trnH* gene was duplicated at the IR regions, but the *trnR* and *trnN* genes were variations. Phylogenetic analyses clarified the interspecific relationship among the *Lycoris* species with whole cp genome sequences. Moreover, we identified the cpSSR sites in different species, together with the highly variable regions identified in interspecific comparison, which could be as useful barcoding markers for the species identification and improve the resolution of phylogenetic relationships within *Lycoris*.

## Figures and Tables

**Figure 1 ijms-21-05729-f001:**
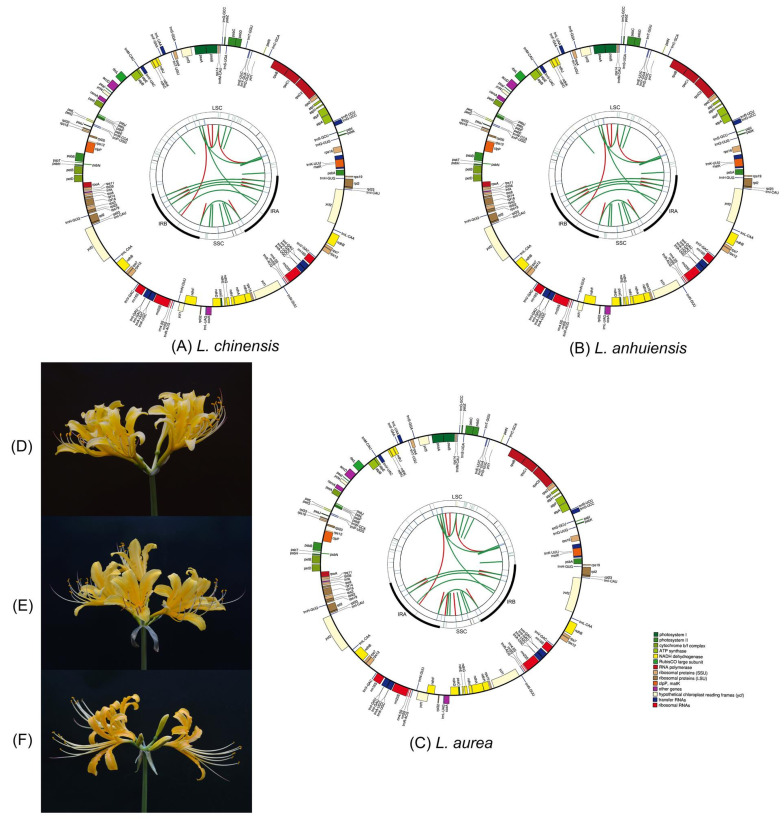
The plastome features and morphological characteristics of *L. chinensis*, *L. anhuiensis*, and *L. aurea*. (**A**,**D**) are the plastome features and morphological characteristics of *L. chinensis.* (**B**,**E**) are *L. anhuiensis*. (**C**,**F**) are *L. aurea.* The schematic map contain four circles from the center going outward, the first circle shows the forward and reverse repeats connected with red and green arcs, respectively. The next circle shows the tandem repeats marked with short bars. The third circle shows the microsatellite sequences. The fourth circle shows the gene structure on the plastome. The genes were colored based on their functional categories. (**A**–**C**) represented the *L. chinensis*, *L. anhuiensis,* and *L. aurea*, respectively.

**Figure 2 ijms-21-05729-f002:**
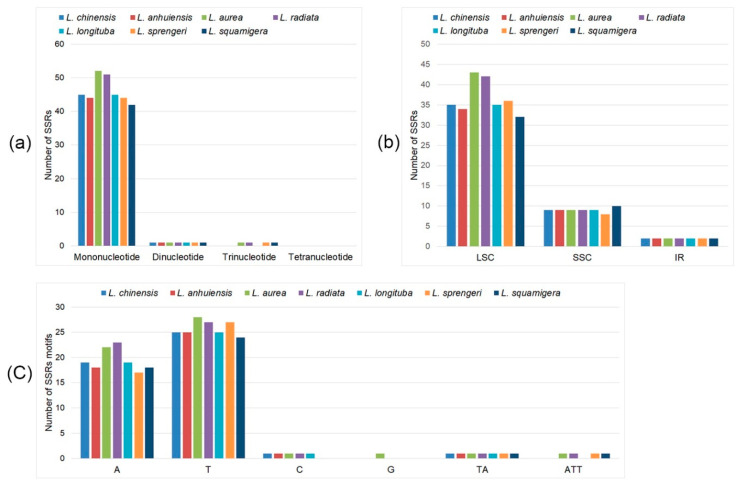
Analyses of simple sequence repeats of plastome in seven *Lycoris* species. (**a**) Numbers of different repeat types; (**b**) Frequency of repeat types in LSC, SSC, and IR regions. (**c**) Numbers of identified SSRs motifs.

**Figure 3 ijms-21-05729-f003:**
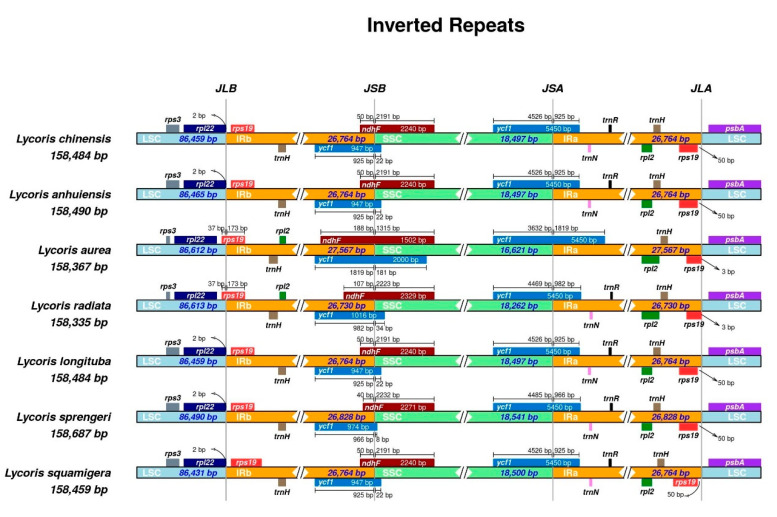
Comparison of boundary distance of large single-copy regions (LSC), small single-copy regions (SSC), and an inverted repeat (IR) among seven *Lycoris* chloroplast genomes.

**Figure 4 ijms-21-05729-f004:**
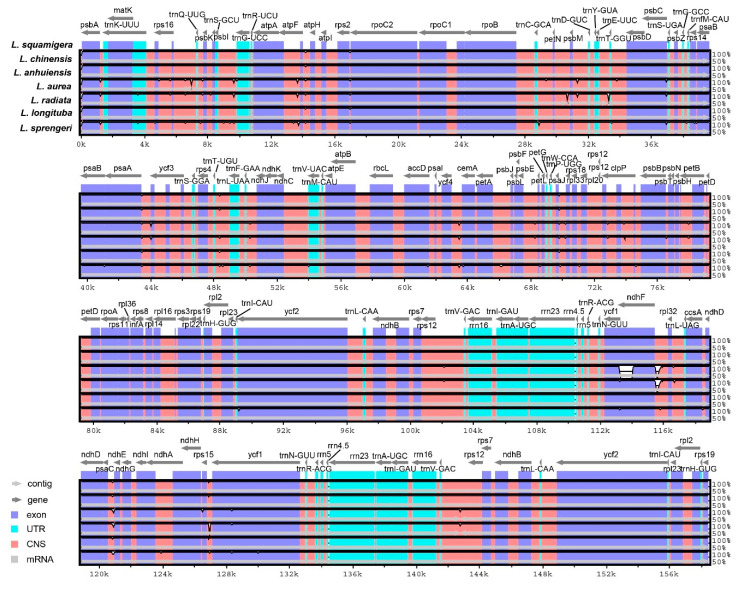
mVISTA identity plot based on the complete cp sequence alignment by Shuffle-LAGAN for seven *Lycoris* species. *L. squamigera* was selected as a reference. The pink regions are Conserved Non-Coding Sequences (CNS), the dark blue regions are exons, and the light-blue regions are UTRs. Arrows signifying genes are drawn above the graphs, pointing in the direction of the gene. 70% cut-off identity was used for the plots, and the Y-scale represents the percent identity from 50% to 100%.

**Figure 5 ijms-21-05729-f005:**
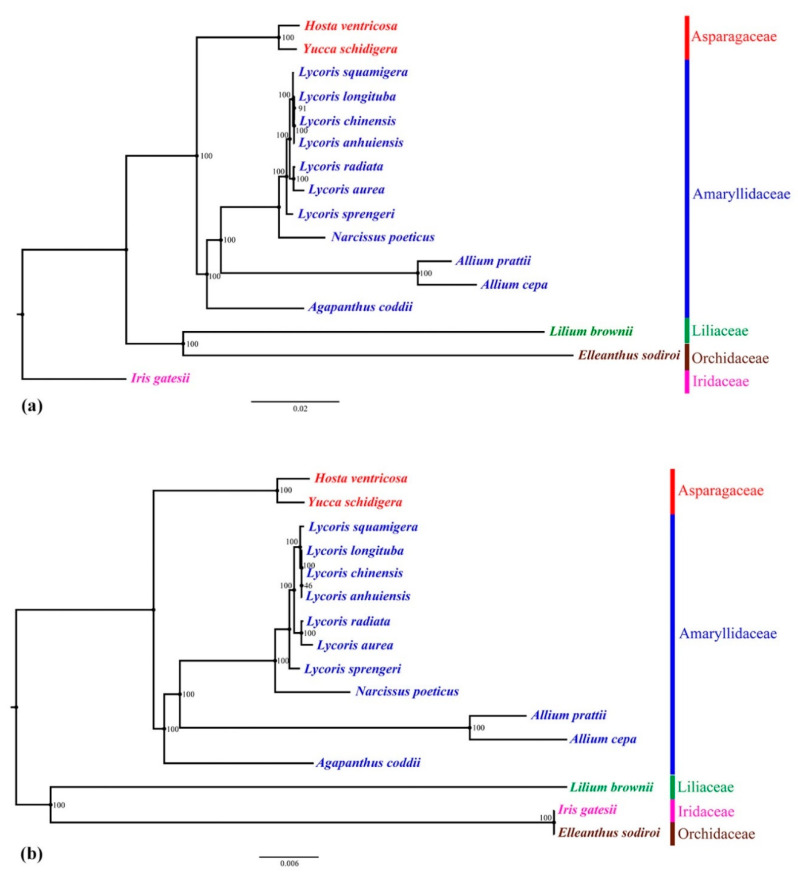
Phylogenetic relationships of the 16 species by maximum likelihood (ML) analyses. (**a**) The topology was constructed by the complete cp genome sequences. (**b**) the tree was constructed based on 72 protein-coding genes.

**Table 1 ijms-21-05729-t001:** Summary of seven chloroplast genomes of *Lycoris* species.

Genome Features	*L. chinensis*	*L. anhuiensis*	*L. aurea*	*L. radiata* [26]	*L. longituba* [27]	*L. sprengeri* [28]	*L. squamigera* [25]
Genome size (bp)	158,484	158,490	158,367	158,335	158,484	158,687	158,482
LSC size (bp)	86,458	86,464	86,611	86,612	86,458	86,489	86,454
SSC size (bp)	18,496	18,496	16,620	18,261	18,496	18,540	18,500
IR size (bp)	26,765	26,765	27,568	26,731	26,765	26,829	26,764
GC content (%)	37.8	37.8	37.8	37.8	37.8	37.8	37.8
No. of genes	137	137	137	137	137	137	159
No. of PCGs	87	87	87	87	87	87	105
No. of tRNAs	42	42	42	42	42	42	46
No. of rRNAs	8	8	8	8	8	8	8

PCGs: protein-coding genes.

**Table 2 ijms-21-05729-t002:** Gene composition of seven *Lycoris* chloroplast genomes.

Category of Genes	Group of Genes	Name of Genes
Genes for photosynthesis	Subunits of photosystem I	*psaA*, *psaB*, *psaC*, *psaI*, *psaJ*
	Subunits of photosystem II	*psbA*, *psbB*, *psbC*, *psbD*, *psbE*, *psbF*, *psbH,**psbI*, *psbJ*, *psbK*, *psbL*, *psbN*, *psbT*, *psbZ*, *ycf3*
	Subunits of NADH-dehydrogenase	*ndhA*, *ndhB* (x2), *ndhC*, *ndhD*, *ndhE*, *ndhF*, *ndhG*, *ndhH*, *ndhI*, *ndhJ*, *ndhK*
	Subunits of cytochrome b/f complex	*petA*, *petB*, *petD*, *petG*, *petL*, *petN*
	Subunits of ATP synthase	*atpA*, *atpB*, *atpE*, *atpF*, *atpH*, *atpI*
	Subunit of rubisco	*rbcL*
Self-replication	Large subunit of ribosome	*rpl2* (x2), *rpl14*, *rpl16*, *rpl20*, *rpl22*, *rpl23* (x2), *rpl32*, *rpl33*, *rpl36*
	DNA dependent RNA polymerase	*rpoA*, *rpoB*, *rpoC1*, *rpoC2*
	Small subunit of ribosome	*rps2*, *rps3*, *rps4*, *rps7* (x2), *rps8*, *rps11*, *rps12* (x2), *rps14*, *rps15*, *rps16*, *rps18*, *rps19* (x2)
	Ribosomal RNAs	*rrn23S* (x2), *rrn16S* (x2), *rrn5S* (x2), *rrn4.5S* (x2)
	Transfer RNAs	*trnA-UGC* (x4), *trnC-GCA*, *trnD-GUC*, *trnE-UUC*, *trnF-GAA*, *trnfM-CAU*, *trnG-GCC*, *trnG-UCC*, *trnH-GUG* (x2), *trnI-CAU* (x2), *trnI-GAU* (x4), *trnK-UUU*, *trnL-CAA* (x2), *trnL-UAA*, *trnL-UAG*, *trnM-CAU*, *trnN-GUU* (x2), *trnP-UGG*, *trnQ-UUG*, *trnR-ACG* (x2), *trnR-UCU, trnS-GCU*, *trnS-GGA*, *trnS-UGA*, *trnT-GGU*, *trnT-UGU*, *trnV-GAC* (x2), *trnV-UAC*, *trnW-CCA*, *trnY-GUA*
Other genes	Subunit of Acetyl-CoA-carboxylase	*accD*
	c-type cytochrome synthesis gene	*ccsA*
	Envelop membrane protein	*cemA*
	Protease	*clpP*
	Translational initiation factor	*infA*
	Maturase	*matK*
Unknown	Conserved open reading frames	*ycf1* (x3), *ycf2* (x2), *ycf4*

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
