# Peer review of "Complete Chloroplast Genomes and Comparative Analyses of L. chinensis, L. anhuiensis, and L. aurea (Amaryllidaceae)"

_ijms, 2020, doi:10.3390/ijms21165729_

Round 1

Reviewer 1 Report

The manuscript entitled “Complete chloroplast genomes and comparative analyses of L. chinensis, L. anhuiensis and L. aurea (Amaryllidaceae)” is well written, the topic of interest and the data provided could be useful to improve knowledge in molecular and genomic characterization of Lycoris.

The paper needs some minor revisions before it can be published:

Lane 84: the sentence “The text continues here.” can probably be removed.

Lane 89: Please add the unit of measurement “bp”; “158,484, 158,490 and 158,367” became “158,484 bp, 158,490 bp and 158,367 bp”

Lane 91: “158480 to 158490 bp” became “158,480 to 158,490 bp”

Author Response

Response to Reviewer 1 Comments

Dear Reviewer,

Thank you very much for your positive comments on our manuscript. We are also much grateful to you for pointing out these mistakes in our manuscript, which are very helpful for the revision of our manuscript. We have carefully revised the manuscript, and they all have been taken into account in the revised manuscript. The revised parts have been clearly highlighted using the “Track Changes”. Below we present the point-by-point responses to your comments.

Point 1: Lane 84: the sentence “The text continues here.” can probably be removed.

Response 1: Thank you very much for pointing out it. I ignored to delete it from the template. I have removed it in the revised manuscript. Please see line 84.

Point 2: Lane 89: Please add the unit of measurement “bp”; “158,484, 158,490 and 158,367” became “158,484 bp, 158,490 bp and 158,367 bp”

Response 2: Thanks for your comments. I have added it in the revised manuscript. Please see line 89.

Point 3: Lane 91: “158480 to 158490 bp” became “158,480 to 158,490 bp”

Response 3: Thanks. I have modified the number format in the revised manuscript. Please see line 91.

Best wishes,

Fengjiao Zhang

On behalf of all the authors

Reviewer 2 Report

The manuscript  describes the results of the analyses based mainly  on the sequencing of the chloroplast genomes of  three species of the genus Lycoris.

Several species of this genus have been shown to contain pharmaceuticaly important alkaloids and there is a lack of detailed phylogenetic studies in this genus.  

While the genomics part has been performed in sufficient extent the phylogenetic part should be largely improved. As the number of the analyzed species is not too high, there is no reason to avoid standard maximum likelihood and Bayesian approaches. The dataset of the presented size can be successfully analyzed using standard personal computer and the necessary programs (e. g, phyml, IQ-TREE, MrBayes,  and  BEAST2) are publicaly available and  free of charge.  The phylogenetic information contained in the analyzed dataset is not completely utilized in analysis when using only distance based methods. Moreover, the authors do not show the obtained lengths of branches in the figure 5, so further loss of information presented to reader occurs. The authors should also consider at least some basic partitioning of the dataset as the model is concerned. The chloroplast is presumed to be inherited as a single molecule but , e.g. coding and non-coding regions probably evolve under different model.

The authors should also try to perform dating of the phylogenetic tree.

In current form, there will be probably low interest of readers but proper phylogenetic analysis can probably improve this point.

I am not a native English speaker but I have also found several typos that could be confusing for the reader.

E.g.:

1) Page 1-line 36: Instead of ",tumor," there should be ",tumor suppressing," or ",anti-tumor,".

2)Page 1-line 36 instead of "..feature is no leaves..." should be probably "...feature is absence of leaves..."

3) Page 1-line 40: Instead of variation there should be probably variety (if obtained by breeding) or possibly variant if it were of spontaneous origin.

The species concerned should be mentioned as well.

4) Page 2 line 55: Instead of "results were conflict" should be "results were conflicting."

5) Page 2 line 88: Instead of "...assembler... was conducted" should be "assembly ... was conducted".

My final conclusion is that the manuscript is acceptable for publication in IJMS

only after major revision which should concentrate mainly on the phylogenetic part.

Author Response

Response to Reviewer 1 Comments

Dear Reviewer,

Thank you very much for your positive comments on our manuscript. We are also much grateful to you for putting forward many good suggestions and pointing out many mistakes in our manuscript, which are very helpful for the revision of our manuscript. In the last week, we have carefully revised the manuscript according to your valuable comments and suggestions, and they all have been taken into account in the revised manuscript. The revised parts have been clearly highlighted using the “Track Changes”. Below we present the point-by-point responses to your comments.

Point 1: The manuscript describes the results of the analyses based mainly on the sequencing of the chloroplast genomes of three species of the genus Lycoris.

Several species of this genus have been shown to contain pharmaceuticaly important alkaloids and there is a lack of detailed phylogenetic studies in this genus.

While the genomics part has been performed in sufficient extent the phylogenetic part should be largely improved. As the number of the analyzed species is not too high, there is no reason to avoid standard maximum likelihood and Bayesian approaches. The dataset of the presented size can be successfully analyzed using standard personal computer and the necessary programs (e. g, phyml, IQ-TREE, MrBayes, and BEAST2) are publicly available and free of charge.  The phylogenetic information contained in the analyzed dataset is not completely utilized in analysis when using only distance based methods. Moreover, the authors do not show the obtained lengths of branches in the figure 5, so further loss of information presented to reader occurs. The authors should also consider at least some basic partitioning of the dataset as the model is concerned. The chloroplast is presumed to be inherited as a single molecule but, e.g. coding and non-coding regions probably evolve under different model.

The authors should also try to perform dating of the phylogenetic tree.

In current form, there will be probably low interest of readers but proper phylogenetic analysis can probably improve this point.

Response 1: Thank you very much for pointing this important point in polygenetic analysis, and we totally agree with you. In the revised manuscript, we constructed the Maximum likelihood (ML) analysis using complete cp genome sequences and 72 common protein coding genes. Considering the common PCGs, we chose 16 species for analysis in the revised manuscript. Two polygenetic trees showed a tiny difference in the relationship among L. chinensis, L. anhuiensis and L. longituba, but both of them suggesting that they were very likely to be derived from one species, and had the same ancestor with L. squamigera. We have clearly described the methods, results, and discussion in the revised manuscript, please see the parts of abstract, 2.5 and 3.4. This major revision made many relative changes in the revised manuscript, please see in lines 27-28, 221-238, 253-257, 293-305, 327, 463-466.

Point 2: I am not a native English speaker but I have also found several typos that could be confusing for the reader.

E.g.:

1) Page 1-line 36: Instead of ",tumor," there should be ",tumor suppressing," or ",anti-tumor,".

2)Page 1-line 36 instead of "..feature is no leaves..." should be probably "...feature is absence of leaves..."

3) Page 1-line 40: Instead of variation there should be probably variety (if obtained by breeding) or possibly variant if it were of spontaneous origin.

The species concerned should be mentioned as well.

4) Page 2 line 55: Instead of "results were conflict" should be "results were conflicting."

5) Page 2 line 88: Instead of "...assembler... was conducted" should be "assembly ... was conducted".

Response 2: Thanks a lot for your objective comment on the language accuracy of our original manuscript. We have revised the points you mentioned and carefully read the manuscript several times and corrected some grammar mistakes. Please see in lines 36, 38, 40, 55, and 87. Other changes were shown in lines 24-25, 105, 108, 199, 212, and 272.

Best wishes,

Fengjiao Zhang

On behalf of all the authors

Round 2

Reviewer 2 Report

The phylogenetic part of the manuscript have been improved so I agree with its publication after minor revisions. Especially there should still be done some corrections in English. I am not a native English speaker but typos in some sentences change their meaning and it can be difficult for a reader to understand the message of the paper. I have found these problems especially in the abstract:

1. The Lycoris is an important medicinal and ornamental plants, which contains about 20 species.

This sentences are not gramatically  correct and meaning is not clear.

Better version could probably be:

The genus Lycoris  (about 20 species) includes  important medicinal and ornamental plants,. Due to the similar morphological features and insufficient genomic resources, germplasm identification and the molecular phylogeny analysis are very limited.

2. Here, we sequenced the complete chloroplast genomes of L. chinensis, L. anhuiensis and L. aurea with similar interspecific morphological traits.

What is the exact meaning of this sentence? Are the species morphologically similar or it means that there is some shared variation is some morphological traits

3.Together with the cpSSR, these variations are useful to develop
molecular markers for germplasm identification.

... these genetic variants... could be better

4. ...were clustered into a monophyletic group, closed to Narcissus and Allium..

The typo changes the meaning . There should be close to Narcissus ...

Author Response

Dear Reviewer,

Thank you very much for your positive comments about our last revised manuscript and more detailed suggestions on our manuscript. We have carefully revised the manuscript and the revised parts have been clearly highlighted using the “Track Changes”. Below we present the point-by-point responses to your comments.

Point 1: The Lycoris is an important medicinal and ornamental plants, which contains about 20 species.

This sentences are not grammatically correct and meaning is not clear.

Better version could probably be:

The genus Lycoris (about 20 species) includes important medicinal and ornamental plants. Due to the similar morphological features and insufficient genomic resources, germplasm identification and the molecular phylogeny analysis are very limited.

Response 1: Thank you for revising the grammar of the sentence, the revised sentence is easier to understand and express meaning clearly. We have corrected in the revised manuscript, please see lines 16-18.

Point 2: Here, we sequenced the complete chloroplast genomes of L. chinensis, L. anhuiensis and L. aurea with similar interspecific morphological traits.

What is the exact meaning of this sentence? Are the species morphologically similar or it means that there is some shared variation is some morphological traits?

Response 2: We are sorry for not expressing the exact meaning of this sentence. Indeed, these three species are morphologically similar, which makes it difficult to identify. We have described it for better understanding in the revised manuscript, please see lines 19-29.

Point 3: Together with the cpSSR, these variations are useful to develop
molecular markers for germplasm identification.

... these genetic variants... could be better

Response 3: Thanks for your good comments, we have revised it in the revised manuscript, please see line 25.

Point 4: ...were clustered into a monophyletic group, closed to Narcissus and Allium..

The typo changes the meaning. There should be close to Narcissus ...

Response 4: Yes, you are exactly correct, the Lycoris species were closed to Narcissus, we have corrected it in the parts of abstract, results and discussion in the revised manuscript, please see lines 27 and 225.